# Reproducibility of Baseline Tumour Metabolic Volume Measurements in Diffuse Large B-Cell Lymphoma: Is There a Superior Method?

**DOI:** 10.3390/metabo11020072

**Published:** 2021-01-26

**Authors:** Florian Eude, Mathieu Nessim Toledano, Pierre Vera, Hervé Tilly, Sorina-Dana Mihailescu, Stéphanie Becker

**Affiliations:** 1Nuclear Medicine Department, Henri Becquerel Cancer Centre, 76038 Rouen, France; florian.eude@chb.unicancer.fr (F.E.); mathieu.toledano@gmail.com (M.N.T.); pierre.vera@chb.unicancer.fr (P.V.); 2QuantiF-LITIS Laboratory (EA 4108-FR CNRS 3638), Faculty of Medicine, University of Rouen, 76130 Mont-Saint-Aignan, France; 3Hematology Department, Henri Becquerel Cancer Centre, 76038 Rouen, France; herve.tilly@chb.unicancer.fr; 4INSERM U1245, Henri Becquerel Cancer Centre, 76038 Rouen, France; 5Department of Statistics and Clinical Research Unit, Henri Becquerel Cancer Centre, 76038 Rouen, France; sorina-dana.mihailescu@chb.unicancer.fr

**Keywords:** positron-emission tomography, large B-Cell lymphoma, metabolic tumour volume, segmentation

## Abstract

The metabolic tumour volume (MTV) is an independent prognostic indicator in diffuse large B-cell lymphoma (DLBCL). However, its measurement is not standardised and is subject to wide variations depending on the method used. This study aimed to compare the reproducibility of MTV measurement as well as the thresholds obtained for each method and their prognostic values. The baseline MTV was measured in 239 consecutive patients treated at Henri Becquerel Centre by two blinded evaluators. Eight methods were compared: 3 absolute (SUV (standardised uptake value) ≥ 2.5; SUV≥ liver SUVmax; SUV≥ PERCIST SUV), 1 percentage SUV threshold method (SUV ≥ 41% SUVmax) and 4 adaptive methods (Daisne, Nestle, Fitting, Black). The intraclass correlation coefficients were excellent, from 0.91 to 0.96, for the absolute SUV methods, Black and Nestle methods, and good for 41% SUVmax, Fitting and Daisne methods (0.82 to 0.88), with a significantly lower variability with absolute methods compared to 41% SUVmax (*p* < 0.04). Thresholds were found to be specific to each segmentation method and ranged from 295 to 552 cm^3^. There was a strong correlation between the MTV and patient prognosis regardless of the segmentation method used (*p* = 0.001 for PFS and OS). The largest inter-observer cut-off variability was observed in the 41% SUVmax method, which resulted in more inter-observer disagreements in the classification of patients between high and low MTV groups. MTV measurements based on absolute SUV criteria were found to be significantly more reproducible than those based on 41% SUVmax criteria. The threshold was specific for each of eight segmentation methods, but all predicted prognosis.

## 1. Introduction

Diffuse large B-cell lymphoma (DLBCL) is the most frequent subtype of lymphoma and accounts for approximately 30 to 40% new cases of lymphoma [1]. Despite improvements in immunochemotherapy, 30 to 40% of patients diagnosed with DLBCL will relapse [2] with a poor prognosis [3,4]. Therefore, it is important to accurately assess the patient’s prognosis so that treatment and monitoring can be adjusted [5]. Prognostic factors of DLBCL are multiple and complementary: clinical, biological, and metabolic. The clinico-biological international prognostic index (IPI) is a powerful prognostic factor recognised in DLBCL. However, it also has limitations, as it does not consider the molecular heterogeneity and biological subtypes of this group of lymphomas, which is reflected in the broad range of clinical outcomes [6].

Functional imaging as ^18^F-fluorodeoxyglucose positron emission tomography/computed tomography (^18^F-FDG PET/CT) has a crucial role in the pre-therapy assessment [7] and evaluation of the response to DLBCL treatment [8,9,10]. Moreover, a combination of the baseline total metabolic tumour volume (MTV) and early response on PET/CT improves progression-free survival (PFS) prediction in DLBCL [11]. The absence of pathological ^18^F-FDG uptake, according to Deauville’s criteria, has an excellent negative predictive value [7]. Consequently, patients with a negative intermediate PET scan have a better prognosis and lower relapse rate compared to those with a positive PET/CT [12,13]. However, this approach does not allow for an a priori assessment of prognosis before the choice of therapeutic management.

The baseline MTV is a recognised independent prognostic factor in DLBCL [14,15,16,17]. It allows for the objective evaluation of the initial tumour burden when performing pre-therapeutic ^18^F-FDG PET/CT. However, it can be challenging to measure. Ideally, its measurement should be conducted in a standardised way to make it as reproducible as possible [18]. In addition, it should also be carried out as quickly as possible to be usable in everyday life, implying the use of automatic, or at the very least, semi-automatic segmentation methods. Recently, automatic delineation using artificial intelligence have been proposed [19,20,21]. From a theoretical point of view, this approach allows a neural network to learn key imaging features in patient and extract tumours automatically while removing physiological uptakes. However, segmentation of tumour burden is complex at both the lesion and whole-body levels. At present, this is still an area of research, and automatic segmentation software are not available in clinical routine to measure MTV in DLBCL. In practice, different segmentation methods have been explored to measure the MTV. There are 2 families schematically. The first, and simplest, uses a fixed standardised uptake value (SUV) threshold beyond which FDG fixation is considered pathological (apart from the physiological fixations and elimination pathways of the tracer). The second, more complex, so-called adaptive method, is based on the use of algorithms that consider the tumour environment [22].

Currently, the most widely used method, especially in clinical studies, is 41% SUVmax. Indeed, Meignan et al. (2014) concluded that the high reproducibility of the MTV determined by the 41% SUVmax method was associated with a good prognosis [23].

Nevertheless, Ilyas et al. (2018) raised the debate again by highlighting the better reproducibility of the 2.5 threshold of the SUVmax, which is a simple, fast, and accessible method [24].

Our study focuses on the reproducibility of different segmentation methods for the measurement of the MTV and its prognostic value by two independent evaluators.

## 2. Results

### 2.1. Patients Chracteristics

We retrospectively enrolled 239 patients with DLBCL. Patients’ clinical characteristics are summarised in Table 1. The mean age (±SD) was 62.8 years (±16.5), 74.5% of patients had a preserved general condition (Eastern Cooperative Oncology Group performance status (ECOG) 0–1), and lactate dehydrogenase (LDH) was increased in almost 70% of patients. The IPI score was high in 58.2% of patients (IPI 3 to 5). The median follow-up was 6.6 years (95% CI: 6.1 to 7.0 years). The five years PFS and overall survival (OS) rates for the entire sample were 54.0 and 60.4%, respectively.

### 2.2. Descriptive Statistics for the MTV Values

For each segmentation method, the mean and its standard deviation, median, first quartile and third quartile were measured for both evaluators (Table 2).

The mean volumes for each segmentation method were as follows: MTV2.5_mean_ = 1017 cm^3^ (evaluator 1) vs. 1023 cm^3^ (evaluator 2); MTV41%_mean_ = 512 vs. 441; MTVLiv_mean_ = 908 vs. 905; MTVPer_mean_ = 906 vs. 899; MTVDai_mean_ = 474 vs. 432; MTVNes_mean_ = 569 vs. 552; MTVFit_mean_ = 623 vs. 547; MTVBla_mean_ = 813 vs. 794.

Bland–Altman plots showed that the largest mean difference between the first and second evaluator in metabolic volumes was with the 41% SUVmax method (71.7 cm^3^) and the Fitting method (76.5 cm^3^) in comparison to <7 cm^3^ for absolute SUV methods and <43 cm^3^ for the other adaptive methods (Figure 1).

### 2.3. Interobserver Variability

The Intraclass coefficient (ICC) varies from 0.96 for the PERCIST method to 0.82 for the 41% SUVmax method (Table 3). There was a significant difference between the ICC obtained with 41% SUVmax and absolute methods with a better reproducibility for absolute methods (*p* = 0.038, 0.023 and 0.038 for SUV ≥ 2.5, Liver SUVmax and PERCIST, respectively).

Likewise, the Daisne method was significantly more reproducible than the 41% SUVmax method (*p* = 0.023) (Table 4).

Kendall’s tau coefficients were 0.93, 0.93, 0.92, 0.89, 0.89, 0.88, 0.86, and 0.85, respectively, for liver SUVmax, PERCIST SUV, SUV 2.5, Black, Nestle, Fitting, Daisne and 41% methods, with a statistically difference between the 41% SUVmax method and all others methods (*p* = 0.01) except Daisne (*p* = 0.7) (Table 3 and Table 4).

### 2.4. Prognostic Value and Survival Analysis

The optimal cut-offs found with the ROC analysis were 552, 295, 487, 486, 340, 396, 352 and 460 cm^3^ for SUV ≥ 2.5, 41% SUVmax, Liver SUVmax, PERCIST, Daisne, Nestle, Fitting and Black, respectively (Table 5). The respective area under the curve for PFS varied from 0.638 to 0.672, suggesting similar performances in term of sensitivity and specificity among the methods (Figure 2).

A high MTV was significantly associated with inferior PFS (*p* = 0.0001) and OS (*p* = 0.0001) with all methods with a 5-year PFS ranging from 65.6 to 69.5% in the low MTV group vs. 37.6 to 42.3% in the high MTV group (Figure 3) and a 5-years OS ranging from 74.6 to 78.6% and 42.2 to 46.7% in the low and high MTV group, respectively (Figure 4).

The largest difference between the evaluator-specific cut-offs (ΔCut-off) was observed for the 41% SUVmax method (Cut-off evaluator 1 = 324 cm^3^ vs. Cut-off evaluator 2 = 252 cm^3^). The difference was ten times smaller for the 2.5 SUVmax method (548 cm^3^ vs. 555 cm^3^) (Table 5).

This variability in the cut-off inter-reader results in a difference in the classification of patients into the low or high MTV groups ranging from 1 to 8 patients depending on the segmentation method used. The 41% method classified 8 patients differently according to evaluator 1 or 2, whereas the 2.5 SUVmax method or Liver SUVmax classified only one patient differently (Table 5 and Figure 5).

Log-rank tests and multivariate analyses were performed using Cox models, including only the 41% SUVmax method for the MTV. The IPI score, the type of chemotherapy and MTV were significantly correlated with PFS and OS (Table 6). These results are in agreement to our previous paper [17].

## 3. Discussion

The powerful independent prognostic value of the MTV is accepted regardless of the segmentation method used [14,15,16,17]. This was also true in the current study for all methods with a continuously increased of risk with MTV for PFS and OS in Cox model.

There was high interobserver agreement for measuring the MTV in all methods, with significantly better reproducibility for the absolute and Daisne methods versus the 41% method.

In our study, Kendall’s Tau was 0.93 for the PERCIST method, 0.92 for SUVmax > 2.5 method and 0.85 for the 41% method compared to 0.96, 0.98, 0.90 respectively in the study by Ilyas et al. [24]. Inter-observer reproducibility was significantly higher for absolute methods compared to the 41% SUVmax method with the same trend in both studies. Kendall’s Tau appeared to be higher on average in the study by Ilyas et al. This is probably due to software differences that allowed for a semi-automatic approach as opposed to a completely manual approach in our case.

Currently, there is no accepted gold standard for assessing the MTV in FDG PET [18]. We will focus more specifically on the two methods most documented in the literature: the 2.5 and 41% methods. In our study, the poorer reproducibility of the 41% method compared to the 2.5 method resulted in a higher variability of the cut-off between the two evaluators. Indeed, the absolute value of the cut-off difference between evaluators in the 41% SUVmax method reached 72 cm^3^ compared to only 7 cm^3^ for the 2.5 SUVmax cut-off method. This difference caused interobserver disagreement for patients with low or high MTV, and therefore, good or poor prognosis concerning 8 patients in the 41% SUVmax method versus only 1 patient in the 2.5 SUVmax method. This could have consequences in patient management when the MTV is used in routine clinical practice or in clinical trials.

At the Henri Becquerel Centre, we have carried out 2 studies partly concerning the same population of patients treated consecutively for DLBCL: Cottereau et al. in 2016 [25] found a threshold at 300 cm^3^ using the 41% method in a population of 81 patients and Toledano et al. in 2018 [17] found a cut-off at 261 cm^3^ using the 41% method in a population of 139 patients, which frames the cut-off value of 295 cm^3^ in the present study. It is accepted that the cut-off is specific to the segmentation method used but also the study population [18,24]. Indeed, the distribution characteristics of the MTV, in particular the median volume, but also age and general state (performance status) are important elements. Indeed, the younger the population and the general state is preserved, the more the cut-off increases.

One of the most comparable studies in terms of patient characteristics is by Song et al. in 2016 [26] which had a population with exclusively advanced stages (100% vs. 78% in our study), equivalent in age (63% > 60 years vs. 64%), and a median MTV2.5 slightly lower (527 cm^3^ vs. 600 cm^3^). They found a threshold at 600 cm^3^ using the 2.5 SUVmax method, relatively close to our cut-off of around 550 cm^3^.

The study by Mikhaeel et al. [11] presented a population with an equivalent median MTV2.5 (595 cm^3^ vs. 600 cm^3^), a lower proportion of advanced stages and younger population (69% advanced stages vs. 78% in our study and 52% < 60 years vs. 36%, respectively). In this study, the cut-off for the 2.5 SUVmax method was around 400 cm^3^ (−28% compared to our cut-off of 552 cm^3^). They also calculated the cut-off of 41% method at 166 cm^3^ (−56% from our 295 cm^3^ cut-off).

Sasanelli et al. in 2014 [14] had a comparable number of advanced stages (82% vs. 78%) but a younger population (31% > 60 years vs. 64%), the method used was the 41% method, the median MTV41% was 315 cm^3^ vs. 284 cm^3^ in our study. The 41% cut-off was 550 cm^3^ versus 295 cm^3^, respectively.

In our study, reproducibility was challenging. Indeed, each lesion was manually contoured using a 3D brush, without automation, blindly. This was probably unfavourable to the 41% method, and probably in part, explains its poorer reproducibility. This method is sensitive to the size of each “box” within which the threshold is applied (via the SUVmax of the hot spot within the box). In our daily experience, it happens quite frequently that the DLBCL diffusely infiltrates an anatomical region, especially in advanced stages, making it difficult to individualise each involved node. There is not always a single way to segment a pathological lesion (Figure 5). This leads to significant inter-observational variation. This has been observed in the cases of patients who were outside the limits of agreement (mean difference ± 1.96 SD) on the Bland–Altman’s figures. This variability does not exist in the case of absolute thresholding that applies to the whole organism for each voxel without the influence of the SUVmax within each initialization box, which probably explains the excellent inter-observer reproducibility (Kendall’s tau ≥ 92%) of the absolute methods. However, the high interobserver reproducibility in the PERCIST and hepatic SUVmax methods is dependent on the determination of the hepatic SUVmean and SUVmax values, which is a prerequisite step in determining a fixed threshold to be applied. In our study, this was measured automatically, so the same threshold was used for both evaluators, eliminating the variability induced by this measurement.

Finally, the Daisne method appears to be an adaptive method that is significantly more reproducible than the 41% SUVmax method. However, it is less accessible in current practice because it is based on a more complex segmentation algorithm. Nevertheless, it is an interesting option to allow better inter-observer reproducibility of the MTV measurement while avoiding the use of a fixed threshold [27].

To our knowledge, this is the largest population of patients with the MTV measured by two readers. This is a consecutive enrolment at the Henri Becquerel Centre. This population is unselected and included all stages of the disease and a range of ages. As there was a median follow-up of 6.6 years (95% CI: 6.1 to 7.0), the PFS and OS data were mature.

Our study had some limitations. This is a retrospective monocentric study on a population that underwent pre-therapeutic PET/CT on an older generation machine (PET/CT Biograph Sensation 16 HiRes). These results should be confirmed for different devices PET/CT and new-generation devices.

More recently, Tout et al. [28] showed that rituximab exposure decreased with an increasing baseline MTV and was found to be predictive of response after induction treatment, OS, and PFS. This also opens perspectives in terms of therapeutic monitoring and personalised medicine. The MTV is a powerful prognostic marker. Its interest is growing, and it could be used in the future as a therapeutic decision tool with a possible intensification in patients with high metabolic volume, and therefore, poor prognosis. However, it is necessary to standardise its measurement through semi-automation so that the least possible intervention by the evaluator is required to make it as reproducible as possible, as defined by Barrington and Meignan [18]. Currently, an area of research is implemented on the theory of repeatable segmentation algorithm independent from the initial input, as reported by Comelli in head and neck and brain tumours [29]. Then, once the measurement method has been defined and an international consensus has been reached, it will be necessary to carry out large multicentre prospective studies to validate the different uses of the MTV in clinical practice. This will also require the training and evaluation of nuclear medicine physicians in its measurement with a benchmark dataset to test their ability to measure the MTV consistently against the expected values.

## 4. Materials and Methods

### 4.1. Patients and Methods

This monocentric study was approved by the Henri Becquerel Centre review board (n°2001B). Patients were informed about the use of anonymised data for the research and their right to oppose this use. The study enrolled consecutive patients between November 2004 and September 2014, who were retrospectively evaluated.

The inclusion criteria were as follows: DLBCL confirmed in all patients by a histopathologic review of the baseline biopsy, treatment using an anthracycline-containing regimen with rituximab; R-CHOP chemotherapy or R-CHOP-like, including R-mini CHOP, R-COPADEMand R-ACVBP, staging with FDG-PET/CT at baseline.

Clinical data obtained from all patients included: sex, age at disease onset, ECOG performance status, extranodal disease, Ann Arbor staging system and the LDH level. This allowed us to calculate the IPI score.

### 4.2. FDG-PET/CT Acquisitions

Images were acquired on a PET/CT Biograph Sensation 16 HiRes (Siemens^®^, Erlangen, Germany) accredited from EARL and performed according to the EANM procedural guidelines [30].

Patients fasted for 6 h and the blood glucose level was <1.7 g/L before injection of the radiotracer. 4.5 MBq/kg of FDG was injected after 30 min of rest. Sixty minutes later (±5 min), acquisitions began with a CT scan in the craniocaudal direction. CT scan parameters were set to 120 kV and 100–150 mAs (based on the patient’s weight) using the dose reduction software (Care-Dose, Siemens Medical Solutions, Hoffman Estates, Knoxville, TN, USA). This yielded a mean effective mA s of 89.1 ± 6.7. The patient’s arms were positioned over his or her head, and the acquisition was performed with free breathing and a 16 × 0.75 mm primary collimation. The duration of the CT scan was 20 s. No contrast media injection was done.

PET image acquisitions immediately followed in the caudocranial direction, and the scan time was based on 3 min per bed position. Six to eight positions were acquired (whole-body); the axial field of view for the 1-bed position was 162 mm with a bed overlap of 25% (plane spacing: 2 mm). The transverse spatial resolution reached 4.4 mm (centred point source in the air). The image matrix was 168 × 168 pixels with 5.3 mm/pixel.

### 4.3. MTV Measurement

All scans were displayed using a fixed SUV scale and colour table. To analyse the interobserver variability, a second nuclear medicine physician (FE), who was blinded, measured the MTV independently from the first observer (MT) using the 8 methods available in the Oncoplanet application (version 3.1; DOSISoft, Cachan, France).

The MTV was computed using the following steps. First, the volumetric regions of interest were placed around each lesion, avoiding physiologic uptake (urinary elimination, heart).

Then the tumour volume was delineated with 8 thresholding methods:-A total of 4 percentage and absolute thresholds: 41% SUVmax (MTV41%) corresponding to volume with counts ≥ 41% of the maximum SUV within individual tumour regions, considered thereafter as the reference [23,30]; SUV ≥ 2.5 (MTV2.5) [11]; SUV ≥ liver SUVmax (MTVLiv); SUV ≥ PERCIST SUV (MTVPer) with PERCIST SUV = 1.5 × (liver mean SUV) + 2 standard deviations [24].-A total of 4 adaptives based on mathematic algorithms: Daisne modified by Vauclin et al. (MTVDai), which iteratively adapts the threshold according to the local signal-to-background ratio [27]; Fitting (MTVFit), which fits the sphere image using a 3-dimensional geometric model based on the spatial resolution in the reconstructed images and on a tumour shape derived from activity thresholding [31]; Nestle (MTVNes) according to the tumour and background intensities [32]; Black (MTVBla) according to the SUVmean [33].

The total metabolic tumour volume was obtained by summing the metabolic volumes of all nodal and extranodal lesions. The bone marrow involvement was included in the volume measurement only if there was focal uptake. The spleen was considered as involved if there was focal uptake or diffuse uptake higher than 150% of the liver background as recommended [18].

The time measurement used to calculate volumes in each method was not carried out due to the preliminary step of depositing boxes on each pathological fixation.

Liver SUVmax and SUVmean measurement were assessed automatically in the right upper lobe of the liver.

### 4.4. Statistical Analysis

The statistical analysis was performed using R software, version 3.6.1 [34]. Agreement between the two observers was evaluated by ICC to measure the consistency between the MTV evaluations and by Kendall’s Tau to measure the rank correlation of the MTV evaluations [35,36]. The 95% confidence intervals (CI) of intraclass coefficient (ICC) and Kendall’s Tau were estimated using 10,000 bootstrap replications with the bootstrap BCa (adjusted bootstrap percentile) [37]. Bland–Altman plots evaluated the means and the differences between the two evaluators, with a 95% CI [38]. The median follow-up was calculated using the reverse Kaplan–Meier method. Overall survival (OS) and progression-free survival (PFS) were estimated from the diagnosis date to death or progression and death, respectively. Since the statistical analyses was carried out after 5 years, data were censored at this time. Survival probabilities were calculated using the Kaplan–Meier method. Log-rank tests and multivariate analyses were performed using Cox models with variable selection prior to analysis according to literature and clinical pertinence. Mean receiver operating characteristic (ROC) curves from the two evaluators were used to predict the PFS at 5 years for each segmentation method by identifying optimal cut-offs [39]. A two-tailed *p*-value below 0.05 was considered statistically significant. For secondary analyses, a Hochberg correction was applied to control the risk of Family-Wise type I error at 5% [40].

## 5. Conclusions

In conclusion, we found that MTV measurements based on absolute SUVmax criteria were significantly more reproducible than those based on 41% SUVmax criteria. The threshold was specific for each of eight segmentation methods, but all predicted prognosis. These results can contribute to setting benchmarks for the measurement of the MTV.

## Figures and Tables

**Figure 1 metabolites-11-00072-f001:**
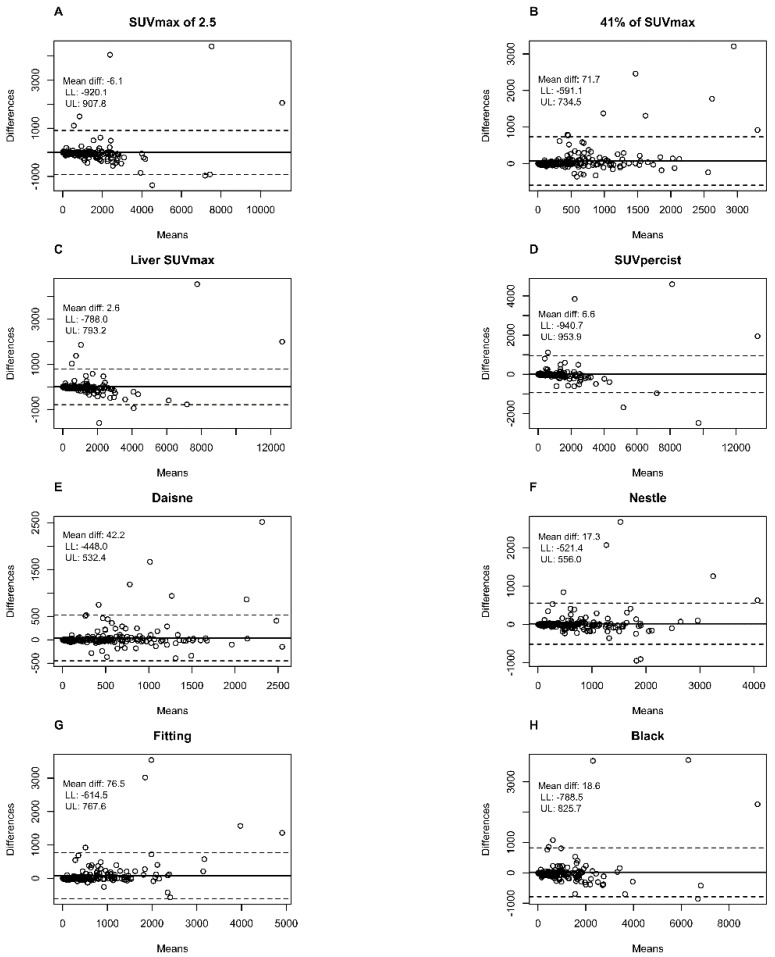
Bland–Altman plots comparing the total metabolic tumour volume measured with the different methods by the two different observers. (**A**) SUVmax of 2.5 method; (**B**) 41% of SUVmax method; (**C**) Liver SUVmax method; (**D**) SUVpercist method; (**E**) Daisne method; (**F**) Nestle method; (**G**) Fitting method; (**H**) Black method. Means: Average metabolic tumour volume of the two evaluators in cm^3^. Differences: Difference in the metabolic tumour volume of the two evaluators in cm^3^. Dotted line: limits of agreement (mean difference ± 1.96 SD); LL: Lower limit 95% confidence interval; UL: Upper limit 95% confidence interval.

**Figure 2 metabolites-11-00072-f002:**
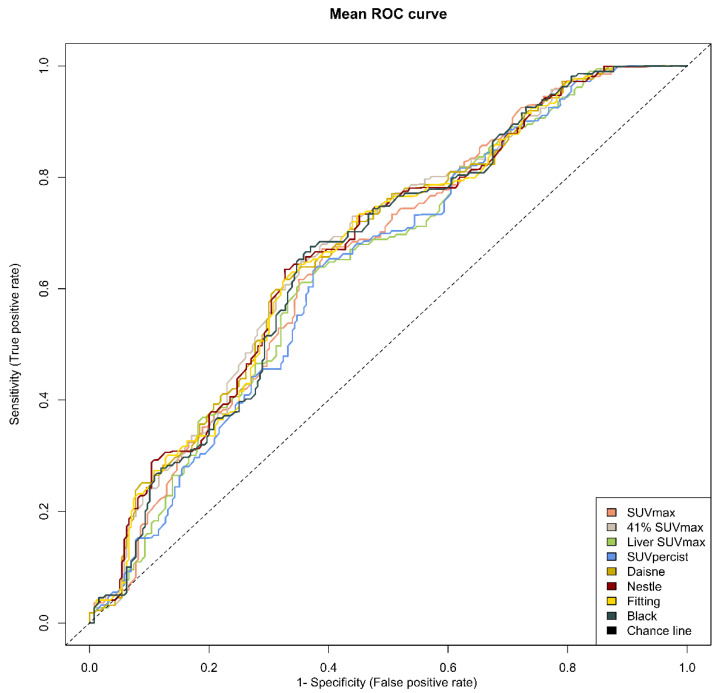
Mean receiver operating characteristic (ROC) curves for each method. Note: ROC curves from the average metabolic tumour volume of the two evaluators for each of the eight methods.

**Figure 3 metabolites-11-00072-f003:**
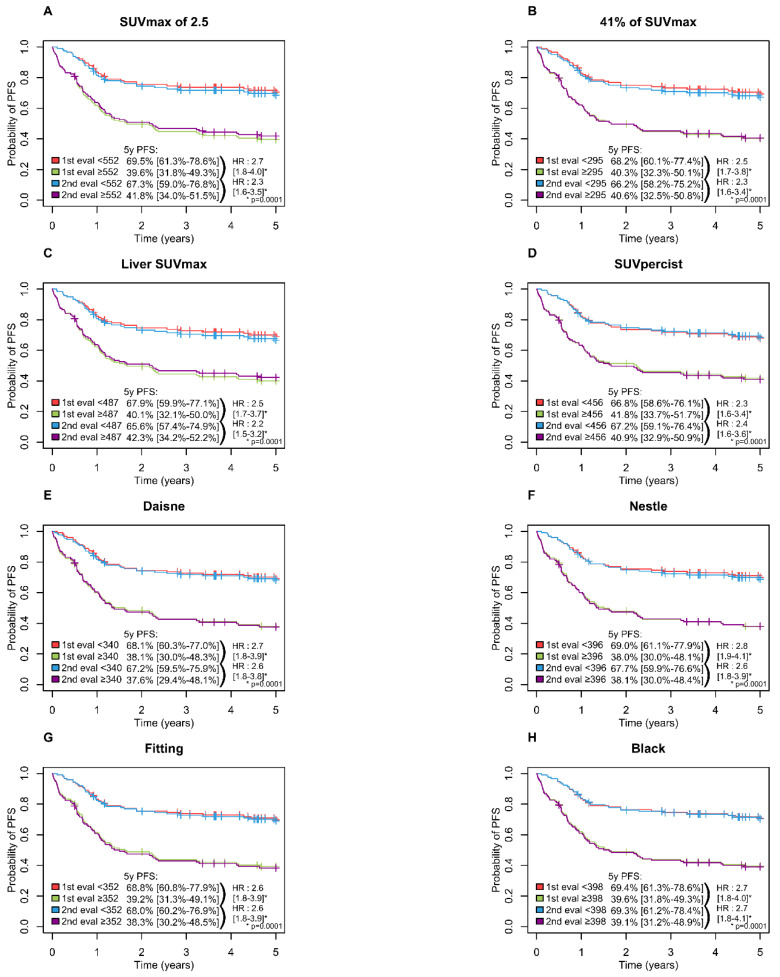
Kaplan-Meier estimates of progression-free survival according to the readers and the eight methods. (**A**) PFS Figure 2. 5 method; (**B**) PFS for 41% of SUVmax method; (**C**) PFS for Liver SUVmax method; (**D**) PFS for SUVpercist method; (**E**) PFS for Daisne method; (**F**) PFS for Nestle method; (**G**) PFS for Fitting method; (**H**) PFS for Black method.

**Figure 4 metabolites-11-00072-f004:**
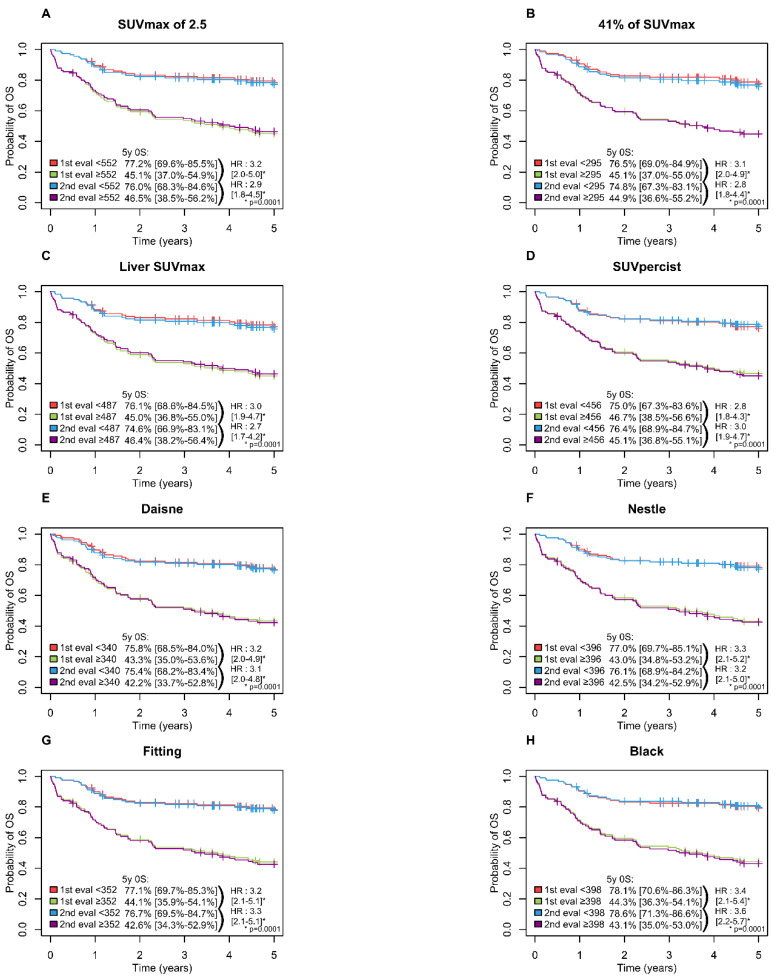
Kaplan–Meier estimates of overall survival according to the readers and the eight methods. (**A**) OS for SUVmax of 2.5 method; (**B**) OS for 41% of SUVmax method; (**C**) OS for Liver SUVmax method; (**D**) OS for SUVpercist method; (**E**) OS for Daisne method; (**F**) OS for Nestle method; (**G**) OS for Fitting method; (**H**) OS for Black method.

**Figure 5 metabolites-11-00072-f005:**
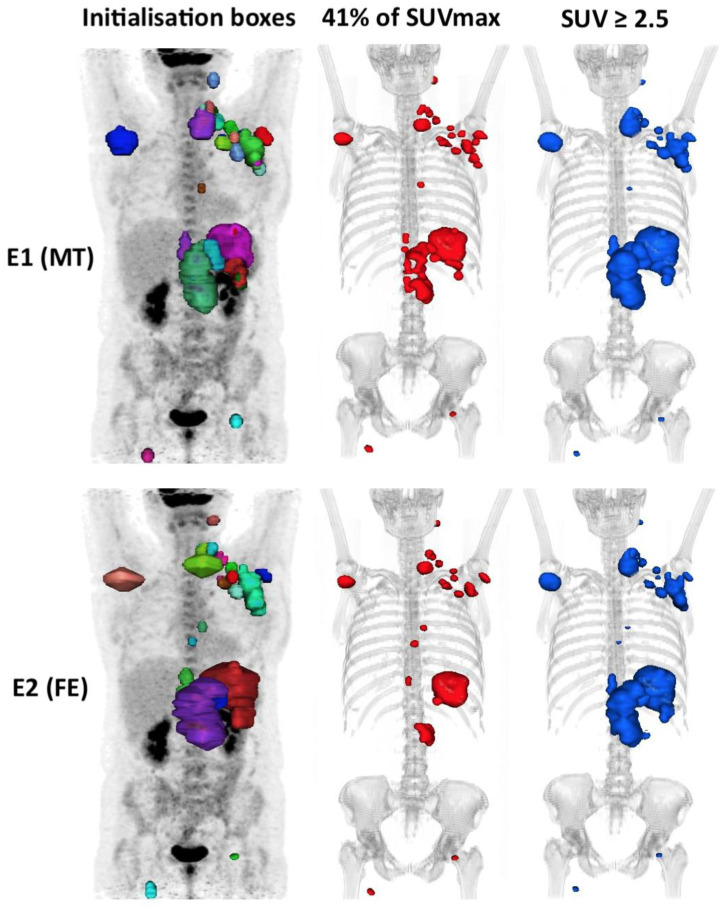
An example of a case outlined using the 41% method and the 2.5 method showing the interobserver variability. In this example, the patient is classified into the high MTV_41%_ by evaluator 1 (324 cm^3^) but low MTV_41%_ by evaluator 2 (241 cm^3^), i.e. Δ MTV_41%_ ≈ 25%, whereas he is classified into the high MTV_2.5_ group for both evaluators with the 2.5 method (respectively, 774 cm^3^ (E1) and 760 cm^3^ (E2), i.e. Δ MTV_2.5_ ≈ 2%).

**Table 1 metabolites-11-00072-t001:** Patient characteristics.

Patients Characteristics	Total (*n* = 239) (%)
Diagnosis age (years), median (min; max)	65.9 (18; 92)
Age ≥ 60 years	152 (63.6)
Female	124 (51.9)
Male	115 (48.1)
ECOG Performance Status (%)	
0	106 (44.4)
1	72 (30.1)
2	35 (14.6)
3	24 (10.0)
4	2 (0.8)
LDH (%)	
Normal	79 (33.1)
Elevated (>480)	160 (66.9)
Ann Arbor stage (%)	
I–II	53 (22.2)
III–IV	186 (77.8)
Extra-nodal sites ≥ 2	155 (64.9)
IPI score (%)	
Low (0–1)	48 (20.0)
Low-intermediate (2)	52 (21.8)
High-intermediate (3)	67 (28.0)
High (4–5)	72 (30.2)
Chemotherapy (%)	
R-ACVBP	67 (28.0)
R-CHOP and others *	172 (72.0)

IPI, International Prognostic Index; LDH, lactate dehydrogenase; ECOG, Eastern Cooperative Oncology Group; R-ACVBP: doxorubicin, vindesine, cyclophosphamide, bleomycin, prednisolone; R-CHOP: Rituximab, cyclophosphamide, doxorubicin, vincristine, prednisolone. * R-miniCHOP, R-COPADEM: methotrexate, cyclophosphamide, vincristine, doxorubicin, prednisolone.

**Table 2 metabolites-11-00072-t002:** Descriptive statistics for the total metabolic tumour volume values.

Method	E	Mean	SD	Min.	Q1 = 25%	Median	Q3 = 75%	Max.
SUVmax 2.5	E1	1017.02	1405.12	1.9	156.61	609.38	1381.71	12117.25
	E2	1023.15	1319.56	4.16	167.92	618.46	1340.28	10065.02
41% SUVmax	E1	512.37	645.57	3.77	80.54	304.57	706.21	4549.41
	E2	440.65	500.27	3.47	74.61	263.73	588.83	2843.38
Liver SUVmax	E1	907.61	1406.49	0.07	105.52	494.39	1317.90	13662.33
	E2	905.03	1302.67	0.08	106.28	478.78	1303.99	11662.33
PERCIST	E1	905.84	1510.22	0.00	87.71	487.00	1208.41	14276.43
	E2	899.24	1457.80	0.00	86.26	445.04	1228.28	12332.37
Daisne	E1	474.19	544.20	2.57	77.42	309.55	678.68	3573.04
	E2	432.00	476.33	2.87	79.28	252.98	594.22	2621.32
Nestle	E1	569.24	666.84	1.99	95.73	359.19	806.33	4383.07
	E2	551.95	624.80	2.61	90.61	324.54	746.83	3753.35
Fitting	E1	623.27	797.44	2.90	104.70	356.58	844.97	5588.33
	E2	546.74	659.45	3.20	93.72	311.25	719.15	4227.48
Black	E1	813.05	1212.58	5.62	118.97	454.26	1085.04	10328.49
	E2	794.46	1099.18	7.17	123.88	414.65	1094.36	8067.13

E: Evaluator 1 (MT) and 2 (FE); SD: Standard Deviation; Q1 and Q3: First and third quartile; SUV: standardised uptake value.

**Table 3 metabolites-11-00072-t003:** ICC and Kendall’s tau between the two evaluators for each segmentation method.

Segmentation Method	ICC (*n* = 239) (95% CI)	Kendall’s Tau (*n* = 239) (95% CI)
SUV ≥ liver SUVmax	0.96 (0.89–0.98)	0.93 (0.87–0.95)
PERCIST SUV	0.95 (0.87–0.98)	0.93 (0.88–0.96)
SUV ≥ 2.5	0.94 (0.85–0.98)	0.92 (0.87–0.94)
Black	0.94 (0.83–0.98)	0.89 (0.84–0.92)
Nestle	0.91 (0.76–0.97)	0.89 (0.83–0.92)
Fitting	0.88 (0.68–0.96)	0.88 (0.83–0.92)
Daisne	0.88 (0.73–0.95)	0.86 (0.80–0.90)
41% of SUVmax	0.82 (0.66–0.92)	0.85 (0.80–0.89)

**Table 4 metabolites-11-00072-t004:** Table of comparison of ICC and Kendall’s Tau by method used.

Methods	SUVmax ≥ 2.5	41% of SUVmax	Liver SUVmax	PERCIST	Daisne	Nestle	Fitting	Black
**SUVmax ≥ 2.5**	-	0.12 (0.03 to 0.27) *p* = 0.038	−0.01 (−0.09 to 0.01) *p* = 0.82	−0.01 (−0.02 to 0.01) *p* = 0.82	0.06 (−0.01 to 0.21) *p* = 0.72	0.03 (0.01 to 0.1) *p* = 0.038	0.06 (0.02 to 0.17) *p* = 0.023	0.01 (−0.01 to 0.02) *p* = 0.82
**41% of SUVmax**	−0.07 (−0.1 to −0.04) *p* = 0.01	-	−0.13 (−0.28 to −0.05) *p* = 0.023	−0.12 (−0.27 to −0.03) *p* = 0.038	−0.06 (−0.10 to −0.03) *p* = 0.023	−0.09 (−0.21 to 0.01) *p* = 0.77	−0.06 (−0.19 to 0.06) *p* = 0.82	−0.11 (−0.27 to −0.02) *p* = 0.102
**Liver SUVmax**	0.01 (−0.01 to 0.02) *p* = 0.74	0.07 (0.05 to 0.12) *p* = 0.01	-	0.01 (−0.01 to 0.06) *p* = 0.82	0.08 (0.01 to 0.23) *p* = 0.072	0.05 (0.01 to 0.17) *p* = 0.144	0.08 (0.02 to 0.25) *p* = 0.023	0.02 (−0.01 to 0.11) *p* = 0.82
**PERCIST**	0.01 (0.01 to 0.03) *p* = 0.07	0.08 (0.05 to 0.13) *p* = 0.01	0.01 (−0.01 to 0.02) *p* = 0.7	-	0.07 (0.01 to 0.22) *p* = 0.42	0.04 (0.01 to 0.12) *p* = 0.038	0.07 (0.02 to 0.2) *p* = 0.023	0.01 (−0.01 to 0.04) *p* = 0.82
**Daisne**	−0.06 (−0.10 to −0.03) *p* = 0.01	0.01 (−0.01 to 0.03) *p* = 0.7	−0.06 (−0.11 to −0.03) *p* = 0.01	−0.07 (−0.11 to −0.04) *p* = 0.01	-	−0.03 (−0.15 to 0.04) *p* = 0.82	−0.01 (−0.11 to 0.11) *p* = 0.99	−0.05 (−0.21 to 0.01) *p* = 0.82
**Nestle**	−0.03 (−0.07 to −0.01) *p* = 0.01	0.03 (0.02 to 0.05) *p* = 0.01	−0.04 (−0.08 to −0.02) *p* = 0.01	−0.05 (−0.09 to −0.03) *p* = 0.01	0.02 (0.01 to 0.04) *p* = 0.01	-	0.03 (0.01 to 0.08) *p* = 0.42	−0.03 (−0.09 to −0.01) *p* = 0.52
**Fitting**	−0.03 (−0.07 to −0.02) *p* = 0.01	0.03 (0.02 to 0.06) *p* = 0.01	−0.04 (−0.08 to −0.02) *p* = 0.01	−0.05 (−0.09 to −0.03) *p* = 0.01	0.02 (0.01 to 0.04) *p* = 0.07	−0.01 (−0.01 to 0.01) *p* = 0.86	-	−0.06 (−0.16 to −0.02) *p* = 0.023
**Black**	−0.02 (−0.06 to −0.01) *p* = 0.01	0.04 (0.02 to 0.08) *p* = 0.01	−0.03 (−0.07 to −0.01) *p* = 0.01	−0.04 (−0.08 to −0.02) *p* = 0.01	0.03 (0.01 to 0.07) *p* = 0.03	0.01 (−0.01 to 0.05) *p* = 0.74	0.01 (−0.01 to 0.04) *p* = 0.74	-

ICC in blue in top right. Kendall’s rate in red on the lower left. *p* value after Hochberg correction.

**Table 5 metabolites-11-00072-t005:** Metabolic tumour volume (MTV) prognostic performance according to the different segmentation methods and its cut-offs.

Method	Se (%)	Sp (%)	Mean AUC	Cut-off (cm^3^)	Cut-off 1 (cm^3^)	Cut-off 2 (cm^3^)	∣ΔCut-off∣(cm^3^)	Low MTV 1 (*n*=)	Low MTV 2 (*n*=)	∣Δlow MTV∣ (*n*=)
SUVmax ≥ 2.5	67.1	61.1	0.655	552	548	555	7	115	114	1
41% of SUVmax	64.8	63.8	0.672	295	324	252	72	117	125	8
Liver SUVmax	63.4	61.9	0.644	487	483	500	17	119	120	1
PERCIST	63.1	62.6	0.638	486	426	465	39	119	125	6
Daisne	61.7	67.3	0.671	340	334	345	11	126	132	6
Nestle	63.5	67.3	0.671	396	398	386	12	123	128	5
Fitting	64.3	64.9	0.667	352	360	335	25	119	126	7
Black	64.8	65.3	0.662	460	379	427	48	121	124	3

Se: Sensitivity; Sp: Specificity. Mean AUC: Mean area under the receiver operating characteristic (ROC) curve. Cut-off: cut-off obtained by the mean ROC curves from the two evaluators. Cut-off 1 or 2: Cut-off determined from the ROC curve of Evaluator 1 or 2. Low MTV 1 or 2: Number of patients in the low MTV patient group (good prognosis) using the evaluator cut-off 1 or 2. ∣Δlow MTV∣: Absolute value of the difference in the patient’s classification into the low MTV group between the evaluators.

**Table 6 metabolites-11-00072-t006:** Cox model (multivariate analysis).

		Disease-Free Survival				Overall Survival			
		Coefficient (SE)	HR (95% CI)	*p*-Value	*p* Value (Hochberg Correction)	Coefficient (SE)	HR (95% CI)	*p* Value	*p* Value (Hochberg Correction)
IPI score	0–2		1	0.02	0.02	1			
	3–5	0.59 (0.24)	1.8			0.66 (0.27)	1.93	0.02	0.02
			(1.12–2.89)				(1.13–3.31)		
CT	ACVBP		1	0.0001	0.0002	1			
	CHOP *	1.11 (0.29)	3.04			1.14 (0.32)	3.12	0.0003	0.0006
			(1.73–5.33)				(1.68–5.83)		
MTV 41%	<295 cm^3^	0.84 (0.22)	1	0.0001	0.0002	1			
	≥295 cm^3^		2.31			1.00 (0.25)	2.72	<0.0001	0.0003
			(1.50–3.55)				(1.68–4.41)		

CHOP *: CHOP and others (R-miniCHOP, R-COPADEM). CT, Chemotherapy; HR, Hazard Ratio; CI, Confidence Interval.

## Data Availability

The data presented in this study are available on request from the corresponding author. The data are not publicly available due to their proprietary nature.

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
