# Peer review of "Reproducibility of Baseline Tumour Metabolic Volume Measurements in Diffuse Large B-Cell Lymphoma: Is There a Superior Method?"

_metabolites, 2021, doi:10.3390/metabo11020072_

Round 1
Reviewer 1 Report
This paper presents results on applying different PET segmentation algorithm to show variations between methods and operators. In general, the paper shows some interesting results. My main critique of the paper is in the presentation of the paper: some important existing approaches are not discussed, and some motivations for the choice of algorithms are not fully clear to me. In my opinion, these issues are all quite possible to address in a revised version. Therefore, my recommendation is ‘Minor revision'. More detailed comments are given below.
- State-of-the art of more recent, advanced, and user-independent segmentation algorithms should be presented in the Introduction section, considering also radiomics studies. For examples, there is some approaches that implement intelligence artificial techniques to identify MTVs. Sample recent papers:
- Stefano, A.; et al. A preliminary PET radiomics study of brain metastases using a fully automatic segmentation method. BMC Bioinformatics 2020, 21, 325.
- Mettler, J.; et al. Metabolic tumor volume for response prediction in advanced-stage hodgkin lymphoma. J. Nucl. Med. 2019.
- Sbei, A.; et al. Gradient-based generation of intermediate images for heterogeneous tumor segmentation within hybrid PET/MRI scans. Comput. Biol. Med. 2020, 119, 103669.
- Comelli, A.; et al. K-nearest neighbor driving active contours to delineate biological tumor volumes. Eng. Appl. Artif. Intell. 2019, 81, 133–144.
Thus, please make a good literature review, compare some results and resubmit the paper.
- In my opinion, a serious segmentation algorithm should be completely independent from the initial input, and therefore, completely repeatable, even when the input is provided by different experts, as reported in many PET segmentation studies. For example, the algorithm presented in the paper: “Development of a new fully three-dimensional methodology for tumors delineation in functional images”. Comput. Biol. Med. 2020, 120, 103701 is a user-independent and three-dimensional algorithm. This is the way to go. Your work indicates precisely this need given the huge differences between the evaluators. It is not important to implement an automatic method but is mandatory to implement a user-independent method as partially reported in rows 326-327. Discuss this point more extensively.
- Algorithms based on SUV threshold are highly affected by the partial volume effect. Discuss this point.
- In addition to MTV, other semi-quantitative parameters can be used to support the prognostic role of FDG-PET based on the semi-quantitative Deauville scale such as in D`Urso, D.; et al. Analysis of Metabolic Parameters Coming from Basal and Interim PET in Hodgkin Lymphoma. Curr. Med. Imaging Rev. 2017, 14, 533–544. Discuss this point.
- 3: Acronyms, such as LDH, ECOG, and other ones, are declared after the first appearance in the text (probably because the Method Section follows that of Results). Fix it.
- Table 1 is not well-formatted. It has some extraneous writing on the right side (rows 91:94).
- Row:352. Report the array size (in voxels and in mm) of PET DICOMs.

Reviewer 2 Report
The authors compared eight methods (three absolute SUVmax methods, one percentage SUV threshoolds method and four adaptive methods) in measuring metabolic tumour volume (MTV) of FDG-PET images of 239 patients with diffuse large B-cell lymphoma at one institute, which was performed by two independent evaluators, and showed that absolute SUVmax methods were significantly more reproducible than the 41% SUVmax method that is most widely used in clinical studies.
Overall, the manuscript is well written, materials and methods are clear and the conclusion is interesting.
The slight concern is whether the same conclusions are applicable for the lesions in brain, where FDG PET is less effective because of high background signals, and also for the comparison in several institutions and multiple cohorts.
Reviewer 3 Report
The authors compared various methods for assessing metabolic tumour volume (MTV) in 239 patients with diffuse large B-cell lymphoma (DLBCL) in one centre. Eight methods were compared: SUV≥2.5; SUV≥ liver SUVmax; SUV≥ mean liver uptake (PERCIST), SUV≥41% SUVmax and 4 adaptive methods (Daisne, Nestle, Fitting, Black).
MTV was predictive of patient prognosis with all segmentation methods. The threshold to be used was, however, specific for a given segmentation method, ranging from 295 to 552 cm3. It is also found that inter-observer cut-off variability was larger with the 41% SUVmax method, which resulted in more inter-observer disagreements in the classification of patients between high and low MTV groups, while MTV segmentation based on absolute SUV methods were more reproducible than.
MTV is gaining importance for prognostic evaluation and the findings are of interest. I have minor commentaries/questions
- There is some discordance between the abstract and MM on how SUV≥ mean liver uptake [PERCIST] is defined, please check.
- There are large variations in median MTV values based on the used technique. Can the authors state which method provides values closer to real measurements. Are there any phantom studies? Are some techniques more robust than others for a wider range of SUV values.
- Was the scoring of the spleen as "involved" or "not" independent of the method used or was there disagreemnt between techniues as regards spleen involvement.
- Why in Table-5 you opted to present differences between cut-off 1 and cut-off 2 as absolute values in cm3 rather than relative difference (%)?
- Table-6 Multivariate analysis: Why you opted to introduce MTV41% in the model as continuous values rather than as binary low vs high MTV (based on the selected cut-off)?
- Also in Table-6 The multivariable model includes Age and IPI. But Age is also part of the IPI. Can these be considered independent variables?
